# Method for airborne measurement of the spatial wind speed distribution above complex terrain

Christian Ingenhorst[1/3], Georg Jacobs[1], Laura Stößel[2], Ralf Schelenz[2], Björn Juretzki[3]

[1] Institute for Machine Elements and Systems Engineering, Aachen, 52062, Germany
[2] Center for Wind Power Drives, RWTH Aachen, Aachen, 52074, Germany
[3] IME Aachen GmbH Institut für Maschinenelemente und Maschinengestaltung, Aachen, 52074, Germany

*Correspondence to*: Christian Ingenhorst (christian.ingenhorst@imse.rwth-aachen.de)

**Abstract.** Wind farm sites within complex terrain are subject to local wind phenomena, which have a relevant impact on a wind turbine's annual energy production. To reduce investment risk, an extensive site evaluation is therefore mandatory. Stationary long-term measurements are supplemented by CFD simulations, which are a commonly used tool to analyse and understand the three-dimensional wind flow above complex terrain. Though being under intensive research, such simulations still show a high sensitivity for various input parameters like terrain, atmosphere and numerical setup. In this paper, a different approach aims to *measure* instead of simulate wind speed deviations above complex terrain by using a flexible, airborne measurement system. An unmanned aerial vehicle is equipped with a standard ultrasonic anemometer. The uncertainty of the system is evaluated against stationary anemometer data at different heights and shows very good agreement, especially in mean wind speed (<0.12 ms$^{-1}$) and mean direction (< 2.4 °) estimation. A test measurement was conducted above a forested and hilly site to analyse the spatial and temporal variability of the wind situation. A position dependent difference in wind speed increase up to 30 % compared to a stationary anemometer is detected.

## 1 Introduction

Complex and mountainous terrain gains importance for wind farm development due to land use conflicts and a high wind potential by speed-up effects at escarpments and steep ridges. Nevertheless, such orographic features as well as obstacles, roughness differences and jet/tunnel effects result in a complex wind field. On these sites, the risk of annual energy production (AEP) overestimation is increased (Lange et al., 2017). Within a wind farm in complex terrain, that was analysed by (Ayala et al., 2017), the AEP of single wind turbines varied up to 25 %, although wake effects seem neglectable when taking into account the park layout and prevailing wind directions.

An increasing demand of renewable energy and high investment risks in case of a false AEP prognosis make wind flows in complex terrain an intensively investigated research topic, concerning both measurement and simulation. Computational Fluid Dynamics (CFD) simulations are a common tool to investigate the spatially distributed wind speeds above complex terrain and is widely used in site assessment and research. Although huge advances in computational power allow even more detailed flow simulations in recent years, CFD simulations still show great sensitivity for assumptions and simplifications such as

terrain details and surface roughness (Jancewicz and Szymanowski, 2017; Lange et al., 2017), atmospheric stability (Koblitz et al., 2014), turbulence models (Tabas et al., 2019) in addition to various numerical parameters. Remaining uncertainties and long computation times make extensive measurements for sites in complex terrain mandatory for a bankable site assessment (International Electrotechnical Commision, 2009; Measnet, 2016; Fördergesellschaft Windenergie und andere Dezentrale Energien, 2017).


Nevertheless, guideline-compliant measurement equipment such as met masts and light detection and ranging (LIDAR) systems are operated stationary with a focus on a maximum statistical coverage. Such systems are not applicable to investigate the spatial deviation of wind speeds within a certain area. State of the art to measure three-dimensional wind fields above complex terrain are multiple doppler LIDAR configurations. Depending on the number of LIDARs, wind speeds in one, two


or three directions can be measured remotely, even at a distance of kilometres. This has been successfully performed in various field studies in complex terrain. For example in Kassel, Germany (Pauscher et al., 2016), triple doppler LIDAR measurements showed good agreement concerning wind speeds in comparison to a sonic anemometer. At Perdigão, scanning LIDARs successfully measured wind speed distributions between a double ridge (Vasiljević et al., 2017). Nevertheless, these measurement systems do have some limitations: as (Stawiarski et al., 2013) point out, the measurement error of a LIDAR


depends, amongst other things, on the angle of the intersecting beams. This can lead to errors "[…] on the order of 0.3 to 0.4 ms$^{-1}$" (Stawiarski et al., 2013). Additionally, multi LIDAR systems do have a significant acquisition cost and take a considerable effort to get erected and operated in steep terrain. Additionally, turbulence intensities measured by multi LIDAR systems still are a topic of ongoing research.

A different approach to measure meteorological variables at specific positions is the usage of an unmanned aerial vehicles


(UAV). Autonomous UAVs, especially fixed-wing systems with pitot-typed wind sensors, have been used for atmospheric research for 20 years (Holland et al., 2001; Spiess et al., 2007; Reuder et al., 2009). In recent years, a fixed-wings system with a 5-hole-probe has been developed to analyse wind speed, inclination angle and turbulence intensity at an escarpment within the swabian alps (Wildmann et al., 2017). In (El Bahlouli et al., 2019), a measurement of a fixed wing system was compared to CFD simulations at the WINSENT test site. Both systems showed plausible results, although the necessary minimum flight


speed of fixed wing systems in general only allows short time measurements for a specific position. Additionally, measurement values also were averaged for a certain flight distance, resulting in an increased probe volume size of several meters. Although both studies aimed to investigate the spatial distribution of wind speeds, temporal changes of the overall wind situation during a single measurement campaign have not been taken into account.

Contrary to fixed wing systems, rotary-wing aircrafts can hold their position mid-air for several minutes. This has three major


benefits: first of all, it allows an easier system validation by just performing hovering flights close to a stationary sensor. This was for example done by (Neumann and Bartholmai, 2015; Palomaki et al., 2017; Nolan et al., 2018) and (Vasiljević et al., 2020), already showing promising results. A further overview is given by (Abichandani et al., 2020), comparing the root mean square error (RMSE) of wind speed and direction measurements of several UAV sensor combinations in literature. So far, turbulence intensity measurements have not been compared yet. The second benefit is, that a stationary, airborne measurement

also allows a reduction of stochastic measurement errors by calculating averaged values for wind speed and direction. Furthermore, rotary-wing UAVs offer greater flexibility concerning their measurement strategy. An exact number, position and duration of measurement points can be chosen. A safe operation at low and high flight levels in complex terrain is also possible. (Shimura et al., 2018) for example use a hexarotor UAV to measure wind vector profiles up to 1000m close to a volcano.

Within our project called *WindLocator*, we have equipped a multi-rotor UAV with a 3D ultrasonic anemometer. In combination with a suitable measurement strategy, we are aiming towards a cost-efficient and accurately measured spatial distribution of wind speed, direction, turbulence intensity and inclination angles. This, finally, would overcome several main limitations of CFD (remaining uncertainties), scanning LIDARs (costs) and fixed wing systems (probe volume size). However, two main challenges have to be overcome within the project before establishing airborne measurement systems as an alternative to

common CFD simulations or LIDAR measurements for investigating complex flow fields:

1. The surrounding air (and its fluctuation) is measured variable, working medium and disturbance for the flying carrier system at the same time. Movements and rotations of the UAV as well as rotor induced flows have a significant impact on the measured wind speed, direction and turbulence intensity. Accuracy of a single measurement point has to be evaluated. In section 2 of this paper, we are going to present the achieved measurement accuracy of the
WindLocator UAV, not only for wind speed and direction, but also for turbulence intensity.

2. CFD Simulations offer the possibility to investigate the 3D wind field at each point for every single time step. UAVs instead measure one point after another and, contrary to scanning LIDARs, take considerable time in doing so. The question arises what kind of measurement strategy is suitable when it comes to merging individual measurement points into one single distribution of meteorological variables. In Section 3, the influence of diurnal wind speed
variation is investigated during two test campaigns above complex terrain, utilizing a simple measurement strategy. Results of the WindLocator are compared to a ground-level anemometer to decide to what extent such a system is suitable as a reference. In the future, those findings combined with a simulation campaign will be used to find a robust measurement strategy.

## 2 Measurement System "WindLocator"

### 2.1 Design

The measurement system, which has been used for the measurement campaigns within this paper, has two main, independent components: a powerful carrier system and a sensor unit, which consists of a commercially available ultrasonic anemometer and a self-developed compensation and data acquisition unit.

The foldable, commercial carrier system is a battery powered octocopter with a flight time of 25 min and a maximum take-off-weight of 12.5 kg. Including the sensor unit, the complete system only weighs 8.5 kg and therefore has a considerable performance reserve. Flights at turbulent air as well as during gust speeds of 25 ms$^{-1}$ have successfully been tested. A real-time-kinematics (RTK) GPS is included to perform high accuracy positional navigation and speed estimation. The open source flight controller has been adapted for an easy setup of specific measurement strategies, which are then autonomously being followed. Although a completely unobserved operation is technically possible, European laws at this moment require an operator to be within sight.

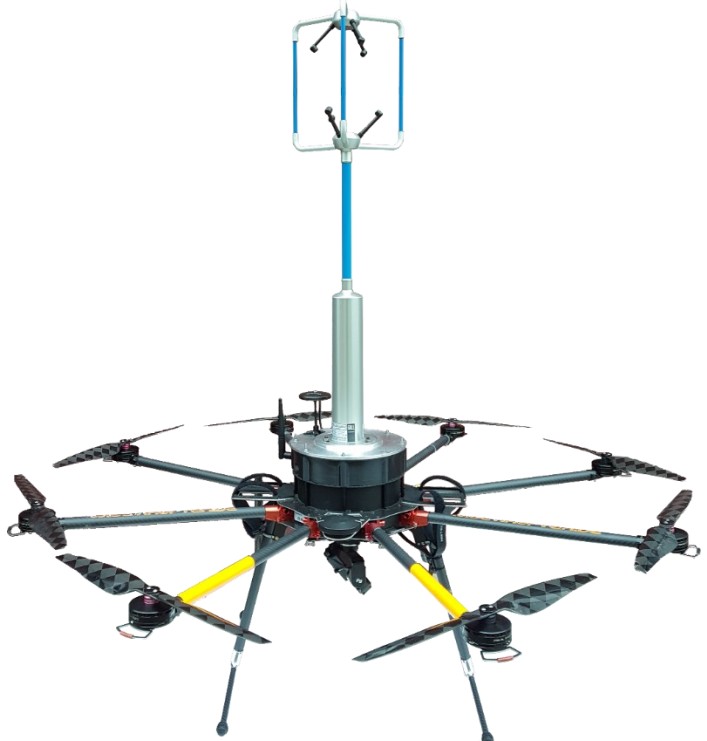

**Figure 1: Measurement system WindLocator (unfolded) without battery packs**

**Table 1 Specifications of carrier system**

| Dimensions | 1060mm (diameter motor-motor), 1250mm (height) |
|---|---|
| Weight (incl. sensor unit) | 8,5 kg |

| | |
|---|---|
| Maximum take-off weight | 12 kg |
| Rotors | 8 x 385mm carbon fibre reinforced polymer rotors |
| Battery | 2 x 10.000 mAh |
| Flight Controller | Pixhawk Cube |
| Flight Times (incl. sensor unit) | ~25 min |
| Air speed | 10 ms$^{-1}$ |

The Gill WindMaster 3D ultrasonic anemometer is placed on top of the compensation unit centred above the rotor plane. Mounting the sensor on top of the UAV has several advantages. First of all, the rotational symmetry of the system allows wind measurement independent from yaw angle and wind direction. Additionally, this setup results in a horizontally centred mass during hovering and therefore leads to relatively small moments to be compensated by the UAV. This improves flight performance and flight time. Aside of that, the downwash above the rotors is less turbulent than below.

The distance of the sensor's measurement volume to the rotor plane is 750 mm and is considered as a trade-off between manoeuvrability and reasonable interaction between wind sensor and propeller induced flows.

**Table 2 Specifications of the ultrasonic anemometer**

| Type | | Gill WindMaster 1590-PK-020 |
|---|---|---|
| Wind Speed | Range | 0-50 ms$^{-1}$ |
| | Resolution | 0.01 ms$^{-1}$ |
| | Accuracy | < 0.18 ms$^{-1}$ |
| Direction | Range | 0-359° |
| | Resolution | 0.1° |
| | Accuracy | 2° @ 12 ms$^{-1}$ |
| Measurement | Internal sample rate | 20 Hz |

Except for the power supply, the self-developed compensation and data acquisition unit is completely independent from the UAV. If requirements concerning the carrier system change, the compensation unit as a whole can be reapplied easily on a new aircraft. It weighs 420 gr and contains all necessary sensors as well as an additional RTK-GPS for an accurate position and speed estimation by means of sensor fusion. Based on analytical calculations and various synthetic experiments, a compensation algorithm was developed, that efficiently reduces measurement errors due to movements of the airborne system as well as its rotors. Additional telemetry transmits measurement data such as wind speeds and directions live to a ground station for in situ analysis. The anemometer data is additionally saved to an internal storage at a rate of 10 Hz.

## 2.2 Validation of the system

All following calculations and measurements have been evaluated based on data that has been processed by the compensation
unit. The system validation in general was conducted on several levels of detail, beginning with the Guide to the Expression
of Uncertainty in Measurement (GUM) to evaluate the standard uncertainty of a single point of measurement. The GUM allows
the calculation of the standard uncertainty without the necessity of a true reference value. Error estimation is done by creating
a mathematical model of the WindLocator, including relevant influences and their uncertainties and combining them into the
system's standard uncertainty, which is +/- 0.37 ms$^{-1}$ in our case.

After several synthetic tests with a fixated UAV to evaluate rotor influences (Figure 2), the WindLocator's compensation unit
was tested during an indoor flight under zero-wind conditions (Figure 3).

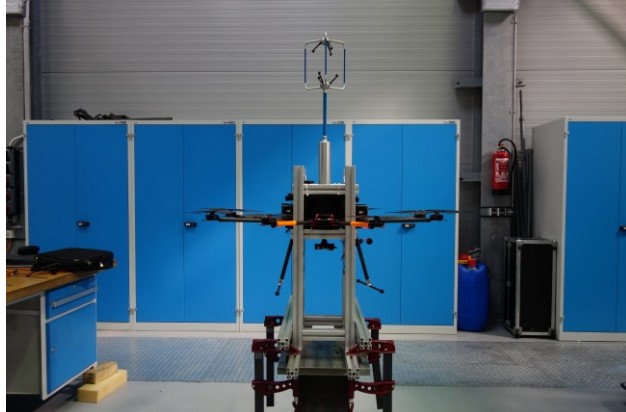

**Figure 2 fixed UAV**

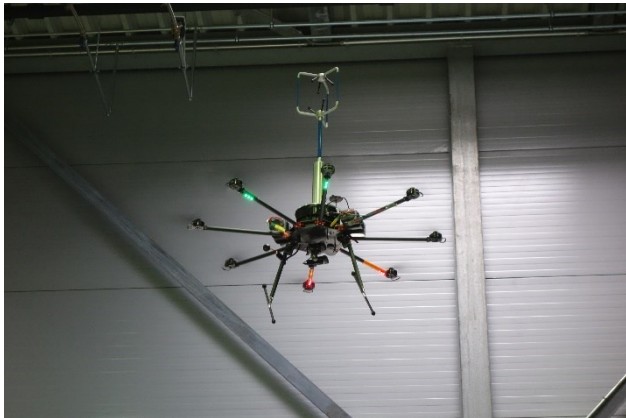

**Figure 3 Indoor flight at zero-wind conditions**

Utilizing the internal barometer an altitude of around 4 m was maintained during our test and pitch and roll axis for minimum
horizontal movements were automatically stabilized. Nevertheless, small sensor inaccuracies made pilot interventions
necessary to remain at sufficient distance to walls. After compensation, the wind data is given out in a global north-east-down
coordinate system and is independent from the specific orientation of the UAV.

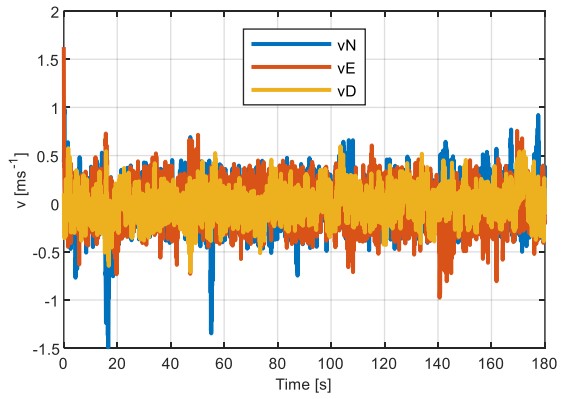

**Figure 4 North-East-Down wind speed components of indoor flight at zero wind conditions**

Figure 4 shows the data in all three measured directions at a resolution of 10 Hz. Peaks, e.g. in vN-direction at second 17 (- 1.5 ms$^{-1}$) and 55 (- 1.31 ms$^{-1}$) are a result of the UAV's horizontal translation due to operator intervention.

**Table 3: Measured wind speed components during indoor flight**

|  | **Wind speed north (vN)** | **Wind speed east (vE)** | **Wind speed down (vD)** |
|---|---|---|---|
| **Mean value [ms$^{-1}$]** | 0.01 | -0.02 | 0.00 |
| **Standard deviation [ms$^{-1}$]** | 0.23 | 0.21 | 0.16 |

As expected, mean wind speeds during the indoor flight are close to zero. Standard deviations up to 0.23 ms$^{-1}$ meet our expectations according to GUM, but clearly show the influence of manual operator control and of the sensor being rather close to the turbulent downwash induced by the rotors.

After proving that under zero-wind conditions, mean values are in good agreement with our expectations, a measurement setup was created to compare the performance of the WindLocator with a stationary anemometer. In flat terrain 2 km west of Aachen (North Rhine-Westphalia, Germany), a stationary anemometer of the same type as the UAV's anemometer was mounted at a height of 3 m above ground level (AGL). Data acquisition and storage for the stationary anemometer were realised at 10 Hz by a self-developed data acquisition system, which uses time stamps synced with an online time server. The UAV time stamps are derived from GPS time signals. The UAV was set to hold position at a height of 3 m. A distance of 4 m to the stationary

anemometer orthogonal to the main wind direction was chosen to avoid interactions of the two measurement systems (Figure 5).

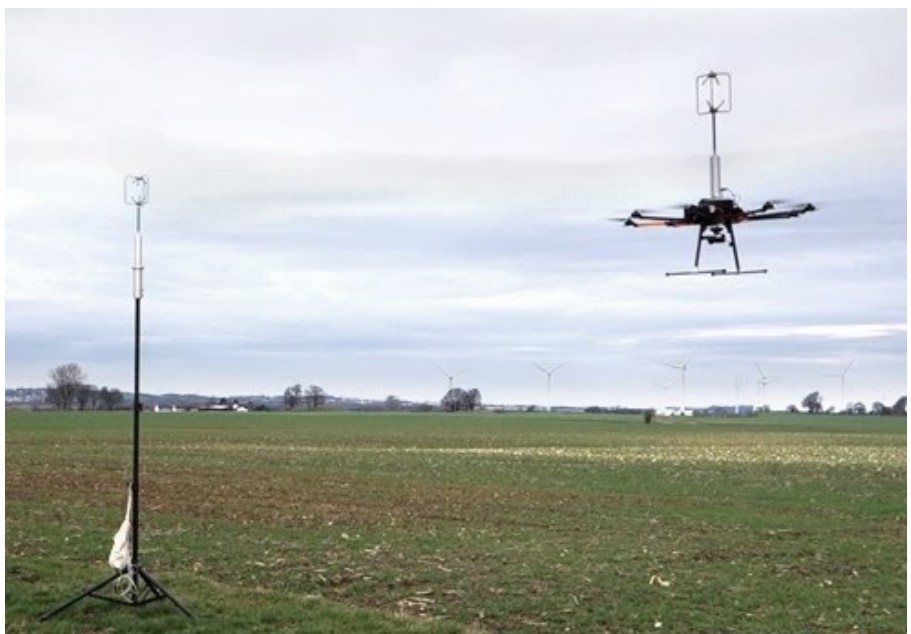

**Figure 5 WindLocator hovering close to stationary anemometer**

Four ten-minute measurements with ~6000 data points each have been conducted, with a short break to switch batteries after the second measurement point. Unlike the indoor tests, all three wind components are combined into a resulting wind speed v for every point of measurement to improve comparability to the stationary anemometer. However, the vertical component $v_D$ in general has a minor impact on the resulting wind speeds.

$$v = \sqrt{v_N^2 + v_E^2 + v_D^2}$$

The following diagram (Figure 6) shows exemplary the compensated wind speeds of the WindLocator in comparison to the stationary reference as well as the corresponding regression plot.

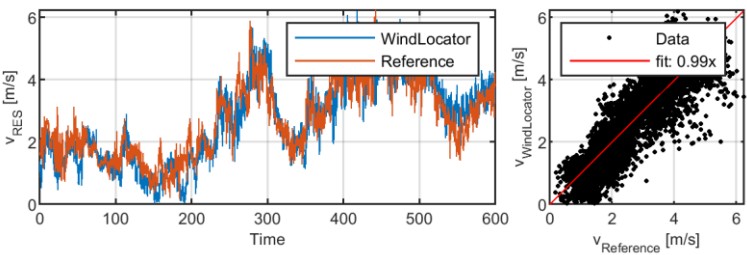

**Figure 6: Resulting ground level wind speed and regression plot of measurement 2**

For all measurement points (see Table 4), a very good agreement of the ten-minute-mean wind speed between WindLocator
and reference has been achieved, especially when taking into account the turbulent wind situation during such a low-altitude flight. Turbulence intensities (TI) up to 44 % have been calculated for the stationary reference. Although there are absolute differences of +1 % (measurement 1) to +6 % (measurement 2 & 4), the WindLocator already provides a good estimation of the prevailing turbulence intensity.

**Table 4 Comparison of measurement points on ground level**

|  | Measurement 1 | Measurement 2 | Measurement 3 | Measurement 4 |
|---|---|---|---|---|
| **Mean speed difference [ms$^{-1}$]** | -0.07 | -0.12 | -0.06 | 0.02 |
| **TI Reference [%]** | 24,5% | 32,2% | 32,2% | 44,3% |
| **TI UAV [%]** | 25,1% | 37,7% | 32,8% | 50,2% |
| **R²** | 0.53 | 0.69 | 0.63 | 0.78 |
| **Standard deviation [ms$^{-1}$]** | 0.58 | 0.55 | 0.58 | 0.64 |


An analysis of wind directions during this experiment was not yet possible, because an accurate orientation of the stationary measurement system could not be guaranteed. This is taken into account for the next experiment at a 134 m met mast under more realistic conditions. The measurement system was tested close to a met mast on a small plateau. Four measurements of 8-10 minutes have been conducted and are compared to the velocity data of a cup anemometer at 134 m and the directional
data of a wind vane at 130 m above ground level. The WindLocator was held on a height of 134 m based on barometer and

GPS data and was then moved closer towards the met mast using the onboard camera system. Because the flight was performed without autopilot, distances to the met mast and exact height vary throughout the measurements (see Table 5). Additionally, that table contains wind speed data analogue to Table 4 as well as information concerning the accuracy of wind direction estimations. For all following calculations, the WindLocator data was averaged to 1 Hz for better comparability to the met mast.

**Table 5: Comparison of measurements on 134m**

| parameter | Measurement 1 | Measurement 2 | Measurement 3 | Measurement 4 |
|---|---|---|---|---|
| Distance to met mast [m] | 26 | 24 | 17 | 16 |
| measurement height [$m_{agl}$] | 134 | 133 | 135 | 135 |
| Mean speed difference [$ms^{-1}$] | -0.21 | 0.01 | 0.20 | -0.06 |
| TI Reference [%] | 15.1 | 18.5 | 11.8 | 15.1 |
| TI UAV [%] | 13.4 | 17.2 | 12.3 | 15.2 |
| $R^2$ | 0.28 | 0.49 | 0.44 | 0.80 |
| Standard deviation [$ms^{-1}$] | 0.88 | 0.88 | 0.63 | 0.47 |
| mean angular difference [°] | 1.2 | -0.7 | 2.0 | 2.4 |
| $R^2$ | 0.56 | 0.49 | 0.82 | 0.71 |
| Standard deviation [°] | 8.5 | 10.8 | 5.1 | 4.6 |

The results during the met mast experiment show a slightly different picture compared to the ground level measurements. The turbulence intensity is still reasonably well estimated. Additionally, measurements 2 and 4 show a good correlation of the WindLocator with the corresponding reference speed. However, mean wind speed deviations for the first and third measurement are not only higher than before, but also vary a lot more compared to the other measurements of that day. Significant deviations mainly occur during the first half of the measurements (Figure 7), e.g. seconds 180 to 270 for measurement 3.

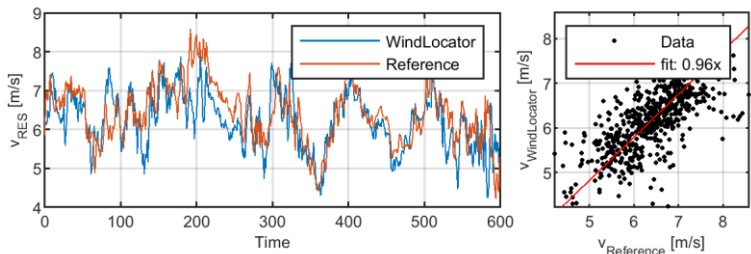

**Figure 7: Resulting wind speed at 134m and regression plot of measurement 3**

Those deviations are a result of the pilot still doing positional adjustments during the measurement point. Nevertheless, those adjustments seem not to have a critical impact on the UAV's wind direction estimation, which shows very good correlation through all measurement points with a maximum mean deviation between met mast and WindLocator of 2.4 °. As an example, absolute wind direction and its regression plot for measurement 3 is shown in Figure 8.

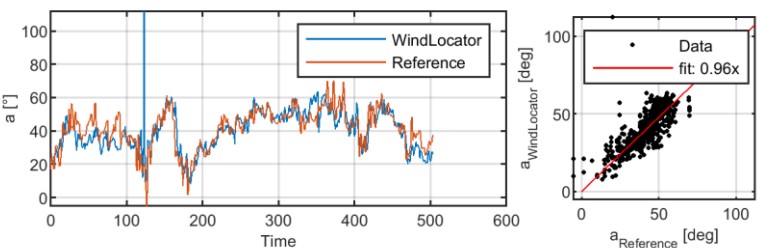

**Figure 8: Wind direction at 134m and regression plot of measurement 3**

The WindLocator performed very well throughout the tests, especially concerning the calculation of averaged measurement quantities like speed and direction. When the system uses its GPS based hover mode without interference by a pilot, mean wind speed differences compared to a reference were below 0.12 ms$^{-1}$ and wind direction differences smaller than 2.4 °. The maximum absolute difference in turbulence intensity was 5.9 % for a high turbulence intensity measurement. Although more measurement points are necessary to finally evaluate the system's performance, initial results in comparison to scanning LIDAR errors seem promising. It also has to be taken into account that the airborne measurement system and a reference cannot measure at the exact same place at the exact same time. Remaining uncertainties always might be a result of spatial deviations in the wind situation, which will be discussed in more detail in the following section.

## 3    Measurement campaign

### 3.1  Test site description

The test site for this measurement strategy is a small hill in the south of North-Rhine-Westphalia in the German Eifel and was chosen for the following reasons:

- With a yearly mean wind speed of 6.5-7 ms$^{-1}$ at 100 m above ground level, the area has rather high wind speeds compared to the rest of the county. Main wind direction is southwest.
- The terrain is considered to be complex. The slope around the hill at most parts is greater than 40 degrees. Forests extend to the south and west of the hill. A small village is located to the northeast, see Figure 9.
- The region in general is easily accessible and was considered suitable for wind turbines.

All diagram coordinates within this chapter are referenced to the UTM coordinate 32U 308450 5604720.

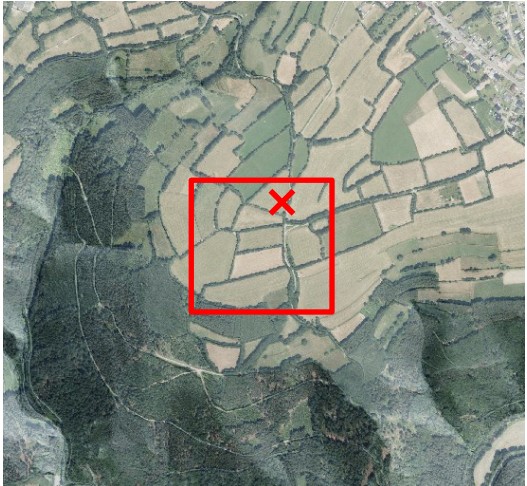

**Figure 9: Test area (□) and stationary measurement location (X) (Source: Geobasis NRW)**

Within this paper, two different measurement campaigns are presented. While the mean wind speeds are within a similar range, wind directions differ resulting in different inflow conditions into the measuring area.

**Table 6: Comparison of measurement campaigns**

|  | Measurement 1 (M1) | Measurement 2 (M2) |
|---|---|---|
| **Mean wind speed at measurement height** | 4.5 ms$^{-1}$ | 4.7 ms$^{-1}$ |
| **Mean wind direction** | 310 ° | 240 ° |
| **Height difference of escarpment in flow direction** | 60 m | 150 m |
| **Surface in upwind direction** | forest and grassland surrounded by sparse hedges | Forrest |

## 3.2 Measurement strategy and evaluation methodology

The presented campaign aims to investigate the feasibility of using a simple measurement strategy for the identification of the spatial distributions of meteorological variables (wind speed, turbulence intensity, inclination) above complex terrain. This information will be used in the further course of the project for the development of the final measurement strategy. The measurement strategy can be described as follows:

- The WindLocator automatically flies to one measuring point after another and measures at each position for a specified duration.
- This duration is chosen as five minutes in the framework of this feasibility study, which is considered to be a reasonable trade-off between limited battery time and statistical coverage for each point.

- To reduce experimental complexity, measurement points are located within a two-dimensional plane. The surveyed plane is roughly 400 m x 400 m and placed on the middle of the test area. All planned measurement points are at the same height above sea level and around 100 m above the lift-off point, see Figure 10.
- At each measurement point, relevant variables like averaged wind speed and direction, turbulence intensity and inclination angle are measured and saved together with the position and a timestamp derived from the GPS.
- Additionally, a ground-level (3 m) anemometer measures wind speed and direction throughout the whole campaign. This ultrasonic anemometer is placed on free grassland surrounded by sparse hedges and captures three-dimensional wind data at 10 Hz.

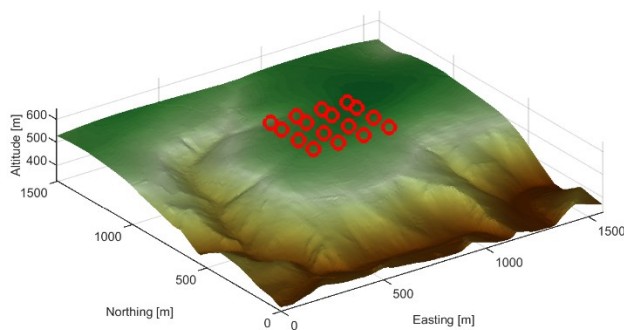

**Figure 10: terrain and measurement points (data source: Geobasis NRW)**

The feasibility study within this paper addresses two basic questions on the postprocessing of the gathered measurement data. In the first step it will be discussed whether the temporal change of the wind speed during the measurement campaign has to be taken into account for the further investigation of the spatial distribution of the meteorological variables. A necessary condition for a constant spatial distribution is a constant wind direction, which will be verified in the beginning. Variations of averaged wind speeds at the stationary reference are used to estimate the impact of temporal variations within the airborne measurements in comparison to expected spatial variations. The result of this analysis is also valid for turbulence intensity, as it depends on the wind speed. Additionally, the spatial distribution of turbulence intensities is checked for plausibility. The influence of temporal changes on inclination angles is checked in a qualitative manner by comparing them to the terrain.

Assuming that the temporal change of the wind speed has a significant effect on the measurement, in a second step it will be investigated whether the ground-level (3 m) anemometer can be used as a reference to compensate temporal changes of the respective variables. Therefore, the ground-level anemometer needs to represent the overall wind situation. This is evaluated using the correlation between ground and airborne measurement data, assuming a linear dependency between those measurements. If correlation is confirmed, wind speed measurements of the WindLocator shall be used to calculate a local speed-up factor in comparison to ground-level wind speed. This distribution then is checked for plausibility.

## 3.3 Results and discussion

Figure 11 gives an overview of the measured resulting wind speeds v from the moving WindLocator and the stationary reference on ground, exemplarily shown for M1. After the data acquisition was started, the UAV heads to the first measurement point, where it is holding position for five minutes 100 m above the start level, before moving on to the next waypoint at the same height. A measured wind speed deviation between WindLocator and reference is expected because of the differences in height and horizontal position of both systems. During the battery swap after four measurement positions, no WindLocator data is available. The stationary reference instead measures non-stop. Measuring 16 points of five minutes, yielding 80 minutes of usable measurement data, has taken around two hours in total.

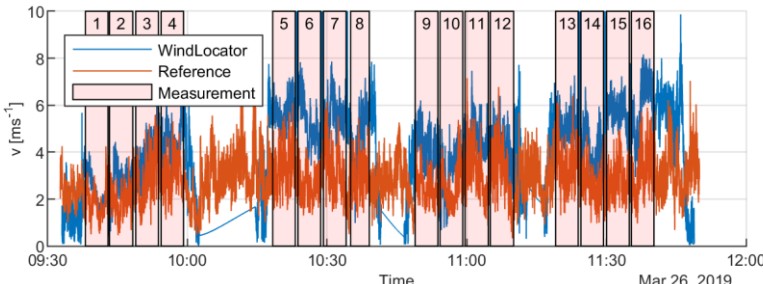

**Figure 11: Comparison of resulting wind speeds from WindLocator (compensated) and Reference at 10Hz for M1**

Figure 12 represents all measurement points for both campaigns, showing the results of the single points measured one after another. During both measurements, wind directions are in good agreement with the mean wind direction. With mean absolute deviations of 9.9 ° (M1) and 11 ° (M2), no significant changes in wind direction during the measurement time of 2 hrs each are found. This validates our assumption, that the distributed wind field will not be influenced by a change in wind direction.

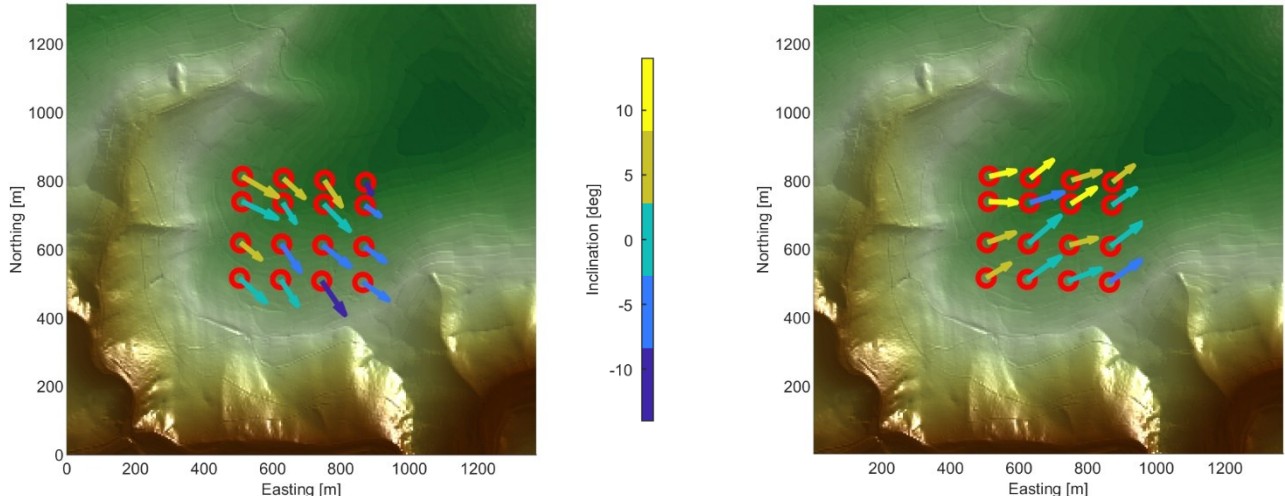

**Figure 12 measurement points, wind vector and inclination angle of M1 (left) and M2 (right)**

Figure 13 shows the averaged wind speeds for one after another measurement point for WindLocator and ground station. Over all UAV measurement points, an absolute variation of mean wind speeds between 2 and 6 ms$^{-1}$ has been detected (Figure 13). As implied earlier, these variations are considered to be too high for spatial deviations due to complex terrain only, especially when taking into account the measurement height of around 100 m. These fluctuations also are a consequence of wind variation over time.

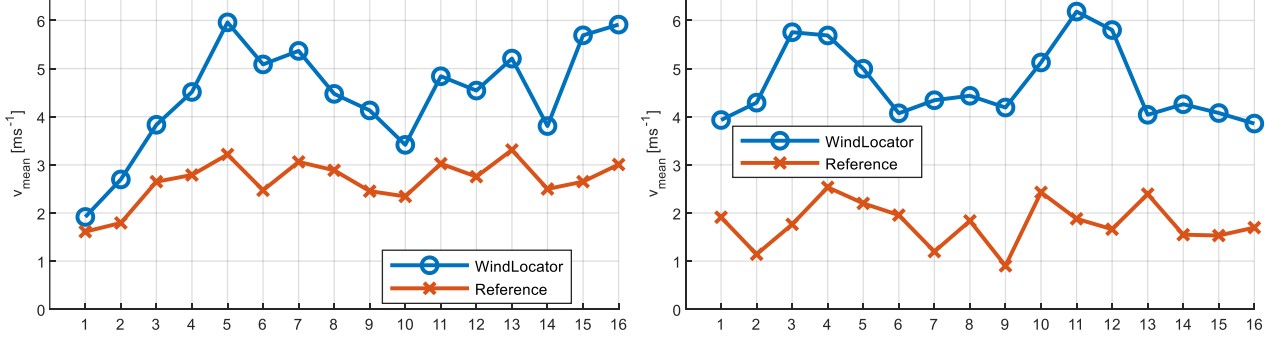

**Figure 13 Mean wind speeds of WindLocator and ground station for M1 (left) and M2 (right)**

Otherwise, the stationary reference (assuming it to be an indicator for the overall wind situation) would not have shown any significant differences in wind speed over time. This is clearly not the case, especially when looking at the normalised reference wind speed, calculated by dividing the mean wind speed value of each point by the maximum mean value of all points of that measurement (Figure 14). Normalised variations of the stationary reference and the WindLocator data are in a comparable order of magnitude (-70 % compared to the maximum wind speed). Because those are comparable and even higher than

expected spatial variations (El Bahlouli et al., 2019; Wildmann et al., 2017) up to 30%, temporal variations clearly have to be compensated for a successful measurement.

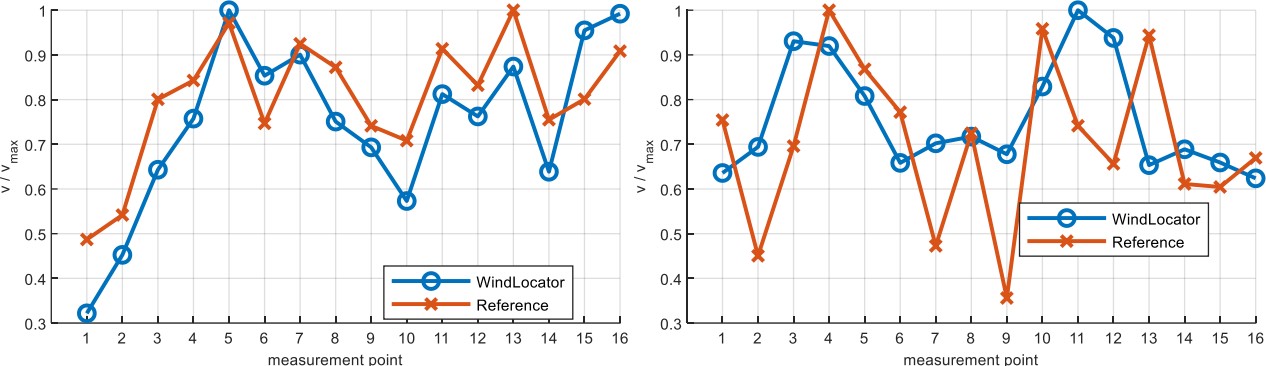

**Figure 14: normalised wind speeds for M1 (left) and M2 (right)**

Figure 15 shows turbulence intensities measured at each single point. The mean turbulence intensity over all measurement points of M2 is 18% and therefore slightly higher than during M1 with 15%. This seems plausible due to the forested and steep escarpment in upwind direction for M2. However, single turbulence intensities within the measured field seem to vary rather strongly (between 10% and 30%) and without obvious influences by terrain and surface. As the normal turbulence model of IEC61400 predicts, turbulence intensity depends significantly on mean wind speed. Very low average speeds of only 2-3 ms$^{-1}$ (see Figure 13 left, Measurement points one and two) might be an explanation for unexpected high turbulence intensity in the north east of M1, for example. Consequently, temporal changes in wind speed have to be taken into account when measuring turbulence intensity distributions.

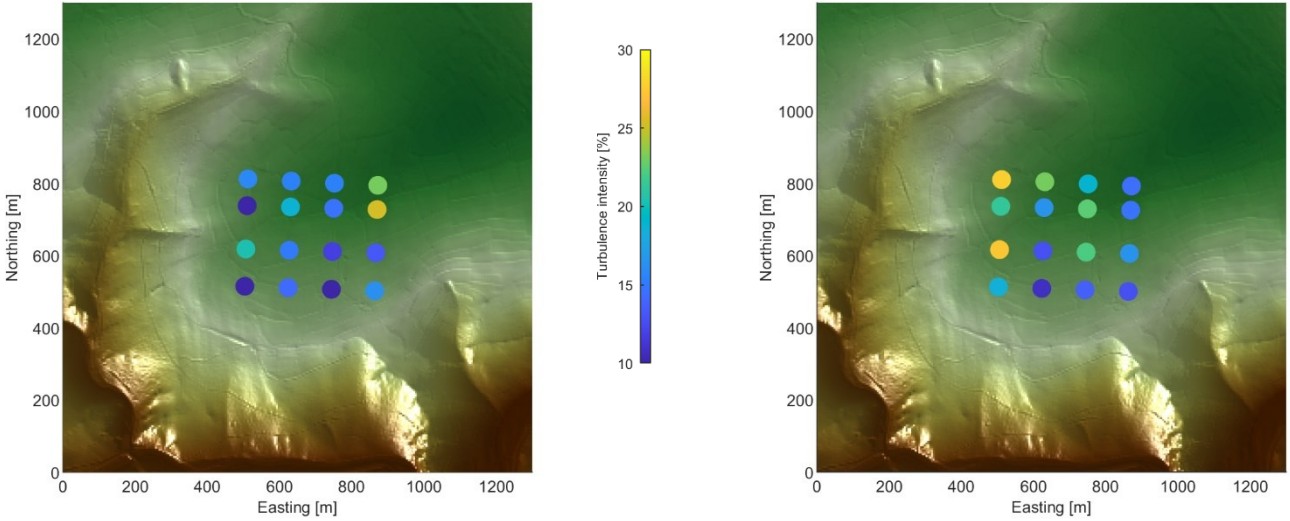

**Figure 15 Turbulence intensities of M1 (left) and M2(right)**

Although wind speeds vary significantly over time, inclination angles do show plausible results (Figure 12). The flow, and therefore the inclination angles, follow the terrain pretty well for M1, varying mostly between +5 ° at the luv side of the hill, switching their sign at the ridge and having -5 ° at the lee side, with a peak of -9 ° at the south close to the escarpment. For M2 on the other side, nearly all angles are above 0, especially in the north with several measured inclination angles higher than 8 °, even up to 12.7 °. Positive inclination angles are considered to be a plausible result from winds passing the steep escarpment in the south west. Temporal variations of wind speed seem to have a minor impact on inclination angle measurements.

All in all, temporal wind speed variations do have a significant impact while measuring a wind speed distribution and therefore have to be compensated. A simple approach would be calculating a wind speed-up value compared to a representative stationary reference. Although the stationary reference in this experiment is only 3 m high, a strong correlation (R=0.86) between relative mean speeds of WindLocator and reference data for M1 is observable, as opposed to M2 (R=0.32). We assume this to be an indicator, that the ground level stationary anemometer for the particular campaign M1 is a suitable reference to also track temporal changes of the overall wind situation. The remaining differences between WindLocator and reference tend to be local wind speed deviations, e.g. due to terrain. Figure 16 shows the speed-up value over time.

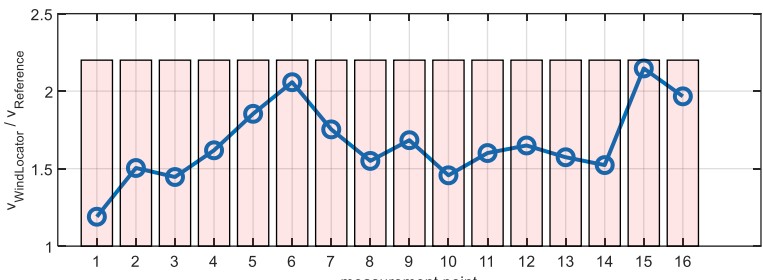

**Figure 16: wind speed increase from reference to WindLocator for M1**

Figure 17 combines the measured results on the one hand and the UAV's GPS data on the other hand to a spatial distribution. The purple arrow indicates the mean wind direction. Each red arrow represents a measurement point, showing the measured horizontal wind direction and indicating with its length the wind speed increase compared to the stationary reference. The wind speed increase then is interpolated linearly between measurement points to create a contour plot. The background shows the digital terrain model data.

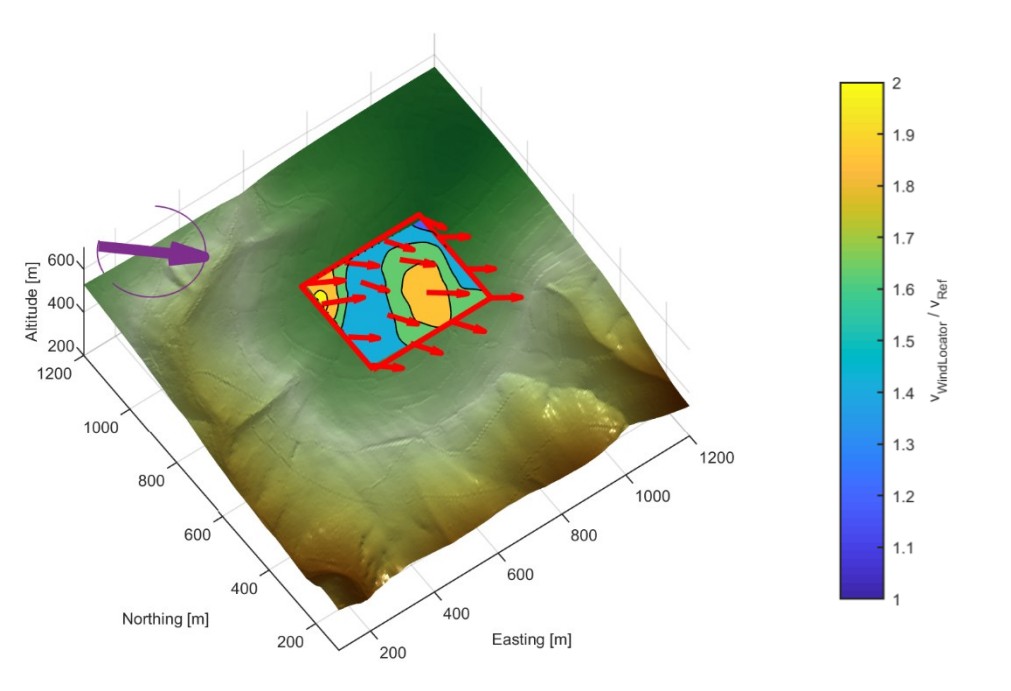

**Figure 17 measured mean wind speed distribution above complex terrain**

The calculated speed-up factor of around two seems plausible, when assuming a logarithmic wind profile with a roughness

315    length of 0.2 m. A variation of the speed-up factor of +30 %/-28 % compared to the mean value is calculated. The highest

increase in wind speed compared to the stationary anemometer is located towards the ridge at the upwind side, which meets

our expectations concerning a speed-up effect at a steep hill. Nevertheless, directly over the highest point, where inclination

angles are close to zero, an unexpected decrease of wind speed-up is detected, followed by an area of higher wind speeds at

negative inclination angles. Towards the plateau in northeast direction, we do see an expected decrease of wind speeds. The

320    results clearly show, that temporal effects must be considered when dealing with turbulence intensities or averaged wind speeds

in general. The simple measurement strategy with a representative ground-level anemometer can be regarded as a proof-of-

concept, leading to an improved estimation of spatial wind deviations for M1 when comparing it to unreferenced data. The

wind speed variations are rather high, but comparable to other campaigns in complex terrain (El Bahlouli et al., 2019;

Wildmann et al., 2017). A plausible explanation for a decrease in wind speed directly on top of the ridge has not yet been

325    found. For the future, a CFD validation shall give insight, whether these effects are a result of measurement errors.

As seen for M2, the presented measurement strategy obviously depends strongly on the stationary reference, its positioning

and expected spatial variations and therefore cannot be considered to be valid in general. A change in wind direction from

310 °(M1) to 240 ° (M2) leads to even lower, less correlated changes in ground level wind speeds with increased turbulence

intensity, presumably as a consequence of surrounding obstacles like hedges. A more robust measurement strategy would probably make use of a more representative stationary reference in greater height.

These findings are currently being evaluated with a simulative approach to find more robust measurement strategies, independent from terrain, location, surface and prevailing wind situation. Once this has been obtained, several of such measurements (for example for different overall wind speeds and directions) might be combined to achieve an extensive insight into the location's wind situation. In future, this could allow bankable site assessment, similar of how CFD simulations are used today.

## 4    Conclusion

Within this paper, a UAV-based measurement system called WindLocator, its validation and its experimental application above complex terrain were presented. The measurement system consists of an octocopter, a commercial ultrasonic anemometer centred above the rotor plane and a self-developed compensation and data acquisition unit. The latter was the enabler to efficiently reduce wind measurement errors due to movements of the UAV and rotor influences. This has been shown in two test scenarios at different wind and turbulence conditions.

At both tests, very good agreement with reference data could be achieved. Mean wind speeds have been estimated with a maximum difference of 0.12 ms$^{-1}$, wind directions with a maximum difference of 2.4 ° during position-controlled hovering. Though rotor influences are a challenge, turbulence intensity estimation was reasonably good. Nevertheless, the compensation unit is under continuous development to improve accuracy at all relevant flight situations.

The biggest advantage of an airborne measurement system is its flexibility, allowing accurate measurements at any arbitrary point in a wind field above any kind of landscape. This could make the WindLocator a potential alternative for CFD simulations in complex terrain, delivering an analogue result for a specific weather situation without long computation times or modelling uncertainties.

During two measurements at a hilly and forested region in the German Eifel, diurnal wind variations were found to be relevant for measuring wind speed distributions and turbulence intensity. Plausible wind direction and inclination were measured even without taking into account temporal variations. Although more advanced measurement strategies are currently under development, for one specific campaign, a simple strategy was sufficient to reduce the influence of diurnal wind speed variations: while the WindLocator automatically was flying from point to point, a stationary reference at ground level was used to compensate the temporal wind speed variations between single measurement points. The result was a plane of four times four measurement points, including information of wind speed increase compared to the reference and three-dimensional wind directions. Spatial differences of approximately +/- 30% compared to a mean value have been found at plausible locations, underlining the necessity of intensive site evaluation in complex terrain. However, this approach significantly depends on how representative the stationary reference is and therefore cannot be considered valid in general.

## Acknowledgements

We thank Windtest Grevenbroich Gmbh for providing validation data from a meteorological mast at their test site.

## Competing interests

Christian Ingenhorst is PhD student at the Institute for Machine Elements and Machine Design (IMSE) and employee at the

IME Aachen GmbH. This company is offering airborne wind measurements as a service.

## Author contributions

Christian Ingenhorst: validation, measurements and analysis

Laura Stößel: measurement site and strategy

Georg Jacobs: supervision of Christian Ingenhorst and paper correction

Ralf Schelenz: supervision of Laura Stößel

Björn Juretzki: supervision of Christian Ingenhorst and paper correction

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
