# Peer review of "Method for airborne measurement of the spatial wind speed distribution above complex terrain"

_Wind Energy Science, 2020_

## Referee Comment (RC1) · Anonymous Referee #1 · 2 Apr 2020

Review of wes-2020-25 Wind speed deviations in complex terrain Christian Ingenhorst et al.

Summary

This manuscript reports on some lab and field tests to check whether a UAV is able to measure the 3D field of turbulence in a wind park of as an alternative of expensive CFD calculations or 3D scanning LIDAR observations. I find this manuscript immature and of marginal scientific impact. See more details below.

Recommendation: reject

Major remarks.

1. Title: The title is not concrete and very broad and does not cover what is discussed in the paper precisely.

2. Research questions: the paper lacks a clear and well-posed research question, or questions and sub questions. As such also the conclusion is rather generally formulated.

3. Methodology: The title suggests this paper is about wind speed deviations/variability. So I do not understand why the paper does not show spectra or wavelet analysis

4. Discussion: the paper also lacks a discussion section that reflects on the strengths and weaknesses of the study, and overall also put the work in context with other studies. Only then the paper can show how it extends the existing knowledge. Also the paper misses a discussion about the representativeness of the atmospheric conditions that were studied.

5. Figures: the paper contains far too many figures. 24 figures is a bizar number, and many of these figures are not essential. Figures 5 and 10 can be removed. I also find that the left panels of figs 6-9 and 11-18 of very limited value, since they are also not much discussed. Figure captions are also not mature and panels have not been labelled a) and b).

Overall I think this work better fits in a scientific report than a journal paper.

Minor remarks: Ln 7: "huge": hyperbolic, should be avoided

Ln 50: USA: not defined

Ln 95: its

Ln 168: corresponding (typo)

Ln 169: speed.

Ln 201: typo in German

---

## Referee Comment (RC2) · Anonymous Referee #2 · 21 Jul 2020

Synopsis: - Authors have proposed and investigated a new method for rapidly characterizing the three dimensional wind field above regions of complex terrain involving the use of an unmanned aerial vehicle, taking measurements of wind speed and wind direction and validating against stationary anemometer data. Initial results are promising, but as suggested by the authors, further work is required to develop a more robust method of compensating the UAV measurements against the stationary reference.

General comments: - While results of the field studies are reasonably well presented, further work is required to interpret the results in the context of the state of the art, including comparisons to other research efforts. More emphasis should be placed on describing how this work contributes to overcoming existing knowledge gaps. Recommendation is for reconsideration after significant revision - Spelling and grammar should be reviewed - some suggestions are provided below but manuscript would benefit from thorough proof-reading.

[Figure]

Specific comments: - You indicate that limtations of the current measurement strategy are too significant to be considered valid (Line 258). It would be useful to describe what criteria are being used to evaluate the validity of the measurement strategy, and to provide additional details on what advancements are believed to be necessary to overcome this issue - Further to the above comment, you mention in Line 32 the notion of bankable site assessments for regions of complex terrain - can you comment on the extent to which UAV-based measurements need to be further developed to meet this benchmark? Is this a desired research outcome? Where do IEC standards fit in with respect to UAV measurements? - Line 201: If possible, it would be useful to indicate the elevation gain from the base of the hill to the peak, as this would give additional context in relation to the measurement plane height of 100m above ground level. It would also be valuable to supply the geographic co-ordinates of the test sites, and the source for the 3D terrain model if applicable - Title of the manuscript could be improved to be more reflective of content, e.g. "Detecting wind speed deviations in complex terrain through airborne measurement" or similar - It would be useful to compare the UAV measurements against CFD and LIDAR studies of the same site; this could possibly be suggested as an area of further work

Figures and tables: - Groups of similar figures (e.g. figs 6-9, 11-14, 15-18) could be combined using indicators a) to d) with a single caption - Figure 19 - would be beneficial to indicate location of stationary ultrasonic anemometer

Typos and spelling/grammar: - Line 31: Suggest replacing "aside of" with "in addition to" if this is the intended meaning - Line 46: The phrase "in opposite" is used here and in several other places in the text. Suggest replacing with "as opposed to" or "contrary to" - Line 52: Suggest replacing "challenges have to be met" with "criteria have to be met" or "challenges have to be overcome" - Lines 54-55: Suggest replacing "but at the same time working medium and disturbance..." with "but is also the working medium and is disturbed by the flying carrier system" - Line 84: Should be "than" not "then" - Line 95: Should be "its" not "it's" - Line 96: Suggest "at a rate" rather than "with a rate"

- Line 105: Suggest "under zero-wind conditions" rather than "at zero-wind conditions" - Line 126: May want to use "orthogonal" or "perpendicular" instead of "rectangular" for improve clarity - Line 168: Should be "corresponding" - Line 169: Period missing after "speed". - Line 210: Suggest avoiding the use of future tense e.g. "going to be investigated" and "plane to be surveyed" and maintain consistency throughout the paper - Line 223: Suggest replacing "Aside of" with "Besides" - Line 231: Suggest replacing "making it" with "yielding" - Line 260: Suggest adding "produce" e.g. "...on the other hand to produce a spatial distribution." - Line 282: The word "to" should be removed: "This could make the WindLocator a potential alternative..."

---

## Author Response (AR1)

We again want to thank Anonymous Referees #1 and #2 for their feedback.

Printed in blue, the author has added some final comments to the already published responses.

**Author's response to Anonymous Referee #1**

1. The title is not concrete and very broad and does not cover what is discussed in the paper precisely.

The authors agree with Anonymous Referee #1. The title might have led to wrong expectations and will therefore be changed in "Method for airborne measurement of the spatial wind speed distribution above complex terrain"

The title has been updated accordingly.

2. Research questions: the paper lacks a clear and well-posed research question, or questions and sub questions. As such also the conclusion is rather generally formulated.

Within the paper, research questions have been presented in form of two specific challenges (lines 54-62) towards an airborne measurement system to investigate the distribution of mean wind speeds above complex terrain. The conclusion concerning the second presented challenge (separation of temporal and spatial effects) will be further refined for the final paper.

Introduction and conclusion have been rewritten to further point out, which knowledge gaps are addressed within the paper. This is also taken into account within "Measurement strategy and evaluation methodology" section, lines 236-250.

**3. Methodology: The title suggests this paper is about wind speed deviations/variability. So I do not understand why the paper does not show spectra or wavelet analysis**

Due to the former, misleading title, the authors understand the expectations of Anonymous Referee #1. Within this paper, we are focusing on the distribution of mean wind speeds at specific positions above complex terrain. 'Deviations' were therefore meant in the spatial and not in the temporal manner. With our application in mind (site evaluation for wind farms) and taking into account the current status of the method, spectral and wavelet analyses are considered to be outside the scope of this paper.

**Spectra and wavelet analyses are still considered to be outside the scope of this paper.**

**4. Discussion: the paper also lacks a discussion section that reflects on the strengths and weaknesses of the study, and overall also put the work in context with other studies. Only then the paper can show how it extends the existing knowledge.**

Strengths and weaknesses of the airborne measurement system have not been discussed within a specific section, but qualitatively throughout the complete paper, for example in context to CFD simulations as well as state-of-the-art measurement equipment. The authors agree, that the paper will benefit from a more detailed assessment of the method at the results chapter, pointing out strengths and weaknesses in a context of other studies. However, the current status of the project does not yet allow a quantitative in-depth validation of the method, which will be part of future publications.

Strength and weaknesses of the method are further described within the introduction and are discussed within the results chapter in a greater extent.

**Also the paper misses a discussion about the representativeness of the atmospheric conditions that were studied.**

The wind speed distribution within this paper is a result of a single, short term (approx. 2hrs) measurement campaign and serves as a proof of concept for the presented method. It is assumed to be representative for the prevailing atmospheric conditions during the campaign, but not for any different weather situations. Therefore, representativeness of the atmospheric conditions was not discussed in detail.

**The results of the method presented within the paper only serve as a proof of concept. Therefore, representativeness was not discussed.**

5. Figures: the paper contains far too many figures. 24 figures is a bizar number, and many of these figures are not essential. Figures 5 and 10 can be removed. I also find that the left panels of figs 6-9 and 11-18 of very limited value, since they are also not much discussed. Figure captions are also not mature and panels have not been labelled a) and b).

Figure 5 was considered to be necessary to enable the reader to evaluate the test conditions. Figure 10 will be removed. The authors agree that figures 6-9, 11-14 and 15-18 could be further reduced to an exemplary plot for each of the following comparisons:

- UAV wind speed measurement to low level anemometer measurement

- UAV wind speed measurement to met mast measurement,

- UAV wind direction measurement to met mast measurement

The left panels (time plots) are considered to be helpful for plausibility, also allowing to point out special events like pilot interaction within the measurement data. Panels will be labelled a) and b) within the final paper.

The number of figures has been reduced accordingly throughout the paper.

**Author's response to Anonymous Referee #2**

**General comments:**

"While results of the field studies are reasonably well presented, further work is required to interpret the results in the context of the state of the art, including comparisons to other research efforts. More emphasis should be placed on describing how this work contributes to overcoming existing knowledge gaps. Recommendation is for reconsideration after significant revision - Spelling and grammar should be reviewed - some suggestions are provided below but manuscript would benefit from thorough proof-reading. "

A comparison to other research efforts, especially CFD and LIDAR with their advantages and disadvantages, has been performed in a qualitative manner throughout the paper. A quantitative in-depth analysis is not yet possible due to the status of our project, but it is planned for future publications.

For the final paper, a more detailed insight on research of measuring wind speed distributions above complex terrain will be given. Additionally, we will explain, that the current project is still ongoing and the paper contains results of the proof-of-concept phase. We will discuss in more detail, which further steps are necessary to raise the UAV's full potential to overcome the presented disadvantages of LIDAR and CFD based wind analysis.

The focus of the paper has been shifted towards the measurement method to better address existing knowledge gaps. A more extensive comparison to existing technologies and research efforts has been added to the introduction (from line 36 to the end of the chapter).

**Specific comments**

"You indicate that limitations of the current measurement strategy are too significant to be considered valid (Line 258). It would be useful to describe what criteria are being used to evaluate the validity of the measurement strategy, and to provide additional details on what advancements are believed to be necessary to overcome this issue. "

The limitations of the current measurement strategy are too significant to be considered valid *in general*. During an ongoing simulation campaign, several measurement strategies, in particular the presented approach, have been evaluated concerning their performance. This is done by virtual test flights within a simulated wind field and shall be part of a future publication. One result is, that the used prototype strategy (single flying measurement system plus single stationary sensor) depends on a reference, which is representative for the area-wide wind situation. We assume this to be the case in the described situation because of the good correlation between normalised wind speeds of the mobile and the stationary system (Fig. 22). In other experimental cases, with differently positioned stationary references and at other wind conditions, we have seen higher variations. The particular strategy therefore is not capable to deliver a plausible wind field without a careful choice of the location of the stationary reference.

For the final paper, this context will be described. Because simulations are still ongoing, more detailed results are not available yet.

A subchapter concerning the used methodology has been added. Additionally, the measurement described above has been added to the results chapter for a more comprehensible and transparent evaluation of the measurement strategy.

"Further to the above comment, you mention in Line 32 the notion of bankable site assessments for regions of complex terrain - can you comment on the extent to which UAV-based

**measurements need to be further developed to meet this benchmark? Is this a desired research outcome? Where do IEC standards fit in with respect to UAV measurements? "**

At the moment, a single airborne measurement can only deliver a "snapshot" of a specific weather situation in terms of wind speed and direction. The results are planned to be used for site assessment in the same way as a single CFD calculation (but without the corresponding modelling uncertainties). However, necessary long-term statistics for a complete UAV-based site assessment can only be realised by a fully autonomous operation, which is not only a technical issue, but also a legal issue in Europe and therefore a mid-term objective.

**A short comment has been added to the paper (lines 329 ff).**

"Line 201: If possible, it would be useful to indicate the elevation gain from the base of the hill to the peak, as this would give additional context in relation to the measurement plane height of 100m above ground level. It would also be valuable to supply the geographic co-ordinates of the test sites, and the source for the 3D terrain model if applicable. "

The elevation gain is roughly 200 m, the 3D model is based on open data from the county of North-Rhine Westphalia. Within the final paper, geographic coordinates and more of the surrounding landscape will be added for additional context.

**More detailed information has been added, see Figure 9, Table 6 and line 212.**

"Title of the manuscript could be improved to be more reflective of content, e.g. "Detecting wind speed deviations in complex terrain through airborne measurement" or similar. "

The authors agree. For the final paper, it shall be changed to "Method for airborne measurement of the spatial wind speed distribution above complex terrain".

**The title has been updated accordingly.**

"It would be useful to compare the UAV measurements against CFD and LIDAR studies of the same site; this could possibly be suggested as an area of further work"

Depending on the LIDAR system, such study usually does not allow to gain insight into the spatial distribution of wind speeds. This specific problem shall be addressed by the UAV based measurement approach. A CFD study nevertheless would be a suitable method for a more in-depth validation of the airborne measurement method, but is not yet in scope due to the status of the project. This shall be addressed in future publications.

**This has been mentioned in line 323.**

**Figures and tables**

Figure titles will be combined, and the location of the ultrasonic anemometer will be added to Fig. 19.

**The position was added in Figure 9.**

**Typos and spelling/grammar**

We want to thank Anonymous Referee #2 for his/her suggestions and take them into account for the final paper.

**The suggestions have been taken into account.**

**Method for airborne measurement of the spatial wind speed distribution above<del>Wind speed deviations in</del> complex terrain**

Christian Ingenhorst1/3, Georg Jacobs1, Laura Stößel2, Ralf Schelenz2, Björn Juretzki3

1Institute for Machine Maschine Elements and Systems Engineering, Aachen, 52062, Germany

2Center for Wind Power Drives, RWTH Aachen, Aachen, 52074, Germany

3 IME Aachen GmbH Institut für Maschinenelemente und Maschinengestaltung, Aachen, 52074, Germany

Correspondence to: Christian Ingenhorst (christian.ingenhorst@imse.rwth-aachen.de)

Abstract. Wind farm sites within complex terrain are subject to local wind phenomena, which have a relevanthuge impact on a wind turbine's annual energy production. To reduce investment risk, an extensive site evaluation is therefore mandatory.
Stationary long-term measurements are supplemented by CFD simulations, which are a commonly used tool to analyse and understand the three-dimensional wind flowflows above complex terrain. Though being under intensive/heavy research, such simulations still show a highhuge 
[revised manuscript text omitted]

---

## Author Response (AR2)

Within this author's response, referee comments are printed in black and our answers in blue. Concrete changes for the revised paper are printed bold.

**Author's response to Anonymous Referee #3 (Report 1)**
We want to thank anonymous referee #3 for his extensive comments.

Ingenhorst et al. present a study about a multicopter equipped with a sonic anemometer to measure wind speed and wind direction at flexible points in space, and in particular in complex terrain. They introduce the system as a possible replacement for site assessment with CFD. Unfortunately I can not see the innovation in this study, since multicopters equipped with sonic anemometers have been reported and multiple times before and the analysis and validation that is done in this study does not go beyond what has been done before.

As we have pointed out, multicopters have been evaluated with various sensors throughout the time concerning wind speed and/or direction. Within our publication, we do show, that even turbulence intensity measurements as well as measurements of wind inclination angles are possible with promising results. Furthermore, we are evaluating accuracies with a focus on typical metrological time scales (10 min) and achieve very good measurement accuracies compared to earlier publications. This underlines and extents the possible metrological applications for UAV based measurements.

One of those applications is the presented measurement of several distributed points of speed and direction and, furthermore, turbulence intensity and inclination above complex terrain. Ultimately, we want to merge these points, measured at different times, into a single wind field. The question arises, whether speed changes throughout the 2h measurement campaign have a relevant impact and to what extent those for example can be compensated with a low-level reference. Based on our knowledge of the relevant literature, this was not yet answered with this application in mind. We consider it to be the first step towards more sophisticated measurement strategies. Nevertheless, we agree as stated inside the paper, that there are still some open questions to be answered until UAV based measurements become a reliable validation tool or even replacement for CFD simulations.

I think the authors have also not done a good job in reviewing the state of the art, because references to very similar systems are missing (e.g. Shimura et al., 2018; Nolan et al., 2018; Reuter et al., 2020; Thielicke et al., 2020, list not complete...).

For sure, literature offers a few more similar UAV based measurement systems since they are around for several years now. We assumed, that Palomaki et al. (IMU based and ultrasonic measurement) and Vasiljevic et al. (LIDAR based measurement) in combination with the overview given by Abichandani et al. allow a representative insight. **We have now added three more comparable systems, although their measurement intention might be different to ours (which is measuring a closed field of wind speed, direction, turbulence intensity and inclination above complex terrain). Shimura et al., who also are already mentioned by Abichandani et al., will now be cited directly because of their comparable measurement purpose (lines 67ff.).**

The publications of Thielicke et al. and Reuter et al., although showing promising results, are still in preprint (initially published months after our initial submission) and therefore not taken into account.

I believe that multicopters as wind measurement systems are a very valuable tool, but I do not at all agree that they can replace CFD in any way and think that suggesting this idea is very misleading for the broad audience. Airborne measurements can be a validation tool for CFD or lidars, but this is not included in this study. I think the authors are missing a good understanding of atmospheric boundary

layer flow if they believe that some short measurement flights can give enough insight for a site assessment, especially in complex terrain. Again, this is reflected in a lack of suitable references and the missing discussion of atmospheric conditions during the measurement campaign.

The measurements of wind speed, direction, turbulence intensity and inclination angle within the paper are examples for two different wind situations above complex terrain. They are the basis to discuss the potential of a ground-level reference to overcome the influence of wind changes throughout the measurement. In-depth discussion of atmospheric conditions was not conducted, because those examples were never mentioned to be representative for the overall yearly wind situation at that site. Neither would two single CFD simulations for different wind directions at that location be sufficient to perform a successful site assessment.

**Within the revised paper, we further explain, that the wind situations are only exemplary measurements. In future, several of such measurements might be combined to achieve a sufficient accurate wind estimation for a site assessment (lines 332 ff.). This will be part of a future publication.**

I believe that the authors have a good instrument for wind measurement, but I strongly suggest that they reconsider what the original scientific contribution is that they can make with this study. I think the development of suitable measurement strategies / flight paths for the analysis of flow structures in complex terrain could be of interest, especially in combination with CFD, but this is not evaluated well enough to be published in WES at this point.

As it is mentioned within the paper, specific measurement strategies and their application will be part of future publications. To this point, we intended to investigate the achievable accuracies of single-point measurements as well as the impact of diurnal changes and whether those can be compensated by a stationary low-level reference

Nolan, P., Pinto, J., González-Rocha, J., Jensen, A., Vezzi, C., Bailey, S., de Boer, G., Diehl, C., Laurence, R., Powers, C., and et al.:Coordinated Unmanned Aircraft System (UAS) and Ground-Based Weather Measurements to Predict Lagrangian Coherent Structures(LCSs), Sensors, 18, 4448, https://doi.org/10.3390/s18124448, http://dx.doi.org/10.3390/s18124448, 201

Reuter, M., Bovensmann, H., Buchwitz, M., Borchardt, J., Krautwurst, S., Gerilowski, K., Lindauer, M., Kubistin, D., and Burrows, J. P.: Development of a small unmanned aircraft system to derive $CO_2$ emissions of anthropogenic point sources, Atmospheric MeasurementTechniques Discussions, 2020, 1–27, https://doi.org/10.5194/amt-2020-234, 2

Shimura, T., Inoue, M., Tsujimoto, H., Sasaki, K., and Iguchi, M.: Estimation of Wind Vector Profile Using a Hexarotor Unmanned AerialVehicle and Its Application to Meteorological Observation up to 1000 m above Surface, Journal of Atmospheric and Oceanic Technology,35, 1621–1631, https://doi.org/10.1175/jtech-d-17-0186.1, 2018.

Thielicke, W., Hübert, W., and Müller, U.: Towards accurate and practical drone-based wind measurements with an ultrasonic anemome-ter, Atmospheric Measurement Techniques Discussions, 2020, 1–29, https://doi.org/10.5194/amt-2020-258, 2020

**Author's response to Anonymous Referee #2 (Report 2)**

We want to thank Anonymous Referee #2 for his positive feedback.

**Author's response to Anonymous Referee #4 (Report 3)**

We want to thank Anonymous Referee #4 for his positive feedback **and have added the suggested technical corrections below to the revised paper.**

Figure 11: Please re-plot with a smaller ordinate range (max of 10m/s)
Figure 15: The color bar legend for TI should not be in %

Typos
L40: this has been successfully performed …
L40: as Stawiarski points out, … - please put a reference not just the name
L80 near end: to decide to what extent such a system

95: which are then autonomously followed.